# Resveratrol Ameliorates Fibrosis in Rheumatoid Arthritis-Associated Interstitial Lung Disease via the Autophagy–Lysosome Pathway

**DOI:** 10.3390/molecules27238475

**Published:** 2022-12-02

**Authors:** Lanxin Bao, Jing Ye, Nannan Liu, Yubao Shao, Wenhao Li, Xuefei Fan, Dahai Zhao, Hongzhi Wang, Xiaoyu Chen

**Affiliations:** 1School of Basic Medical Sciences, Anhui Medical University, Hefei 230601, China; 2Department of Respiratory Medicine, The Second Affiliated Hospital of Anhui Medical University, Hefei 230601, China; 3Microscopic Morphological Center Laboratory, Anhui Medical University, Hefei 230032, China; 4Department of Clinical Medicine, Anhui Medical University, Hefei 230032, China; 5Institute of Health and Medical Technology, Hefei Institutes of Physical Science, Chinese Academy of Sciences, Hefei 230031, China; 6Hefei Cancer Hospital, Chinese Academy of Sciences, Hefei 230031, China

**Keywords:** RA-ILD, resveratrol, autophagy, fibrosis, P62/SQSTM1

## Abstract

Interstitial lung disease associated with rheumatoid arthritis (RA-ILD) can lead to interstitial fibrosis and even lung failure as a complication of rheumatoid arthritis (RA), and there is currently no effective treatment and related basic research. Studies have found that resveratrol (Res) can improve the progression of RA by regulating autophagy, and increasing evidence supports the connection between autophagy and common interstitial lung disease (ILD). We explored changes in autophagy levels in fibrotic lungs in RA-ILD and found that the level of autophagy is enhanced in the early stage but inhibited in the late stage. However, resveratrol treatment improved the level of autophagy and reversed the inhibition of autophagy, and attenuated fibrosis. We created corresponding cell models that exhibited the same phenotypic changes as animal models; under the effect of resveratrol, the level of fibrosis changed accordingly, and the fusion process of lysosomes and autophagosomes in autophagy was liberated from the inhibition state. Resveratrol effects were reversed by the addition of the late autophagy inhibitor chloroquine. These results suggest that resveratrol attenuates pulmonary fibrosis, increases autophagic flux, and modulates the autophagy–lysosome pathway, and particularly it may work by improving the formation of autophagic lysosomes, which may be an effective treatment for induced RA-ILD.

## 1. Introduction

Interstitial lung disease associated with rheumatoid arthritis (RA-ILD) has received much attention in recent years. Rheumatoid arthritis (RA) is a common autoimmune disease that causes progressive articular damage, functional loss, and comorbidities [1]. The most common extra-articular manifestation of RA is interstitial lung disease (ILD) [2]. ILD is a group of diseases characterized by interstitial fibrosis and inflammation. Clinically, the most common type is the usual interstitial pneumonitis (UIP), which can rapidly deteriorate lung function and even lead to lung failure [3,4]. For UIP in ILD, its pathology is characterized by extensive bilateral pulmonary interstitial fibrosis [5]. Since ILD is a common manifestation and a poor prognostic factor in RA, the management of RA-ILD often directly affects the prognosis of RA patients. Currently, RA needs to be stabilized immediately as the initial treatment management for the chronic course of RA-ILD. In patients with progressive RA-ILD and those at high risk of infection or with fibrotic predominance, clinicians should consider initiating antifibrotic medications [6]. However, it has been demonstrated that beneficial drugs commonly used to treat arthritis, such as methotrexate, leflunomide (LEF), and anti-tumor necrosis factor-alpha, do not alleviate or even worsen the development of lung disease. Some drugs may provide hope for the treatment of RA-ILD, including immunomodulators, such as mycophenolate and rituximab, and newly studied antifibrotic agents [7]. However, there is still little high-quality evidence to guide the treatment of RA-ILD [8,9].

Autophagy is a self-digestion process that occurs in eukaryotic cells and mediates several intracellular biological functions, such as the removal of cytoplasmic material [10]. Autophagy (herein referred to as macro-autophagy) is mediated by an autophagosome, a process in which a part of the cytoplasm is surrounded by an isolation membrane or phage to form an autophagosome, which is degraded by lysosomal enzymes in the autophagosome by fusion with the lysosome. It consists of several steps, including the formation of phagosomes and autophagosomes in the early stage, the fusion of autophagosomes and lysosomes to form autophagolysosomes, and the degradation of autophagolysosomes later [10,11]. Autophagy serves primarily as an adaptive function to protect organisms from a variety of diseases. It selectively targets dysfunctional organelles, intracellular microorganisms, and disease-causing proteins, and defects in these processes may contribute to the development of many diseases [12,13]. It has been proven that autophagy plays a crucial role in maintaining the metabolic homeostasis of lung tissue and the occurrence and development of chronic respiratory diseases and is involved in regulating a variety of biological functions in the lung, such as the inflammatory response, DNA damage repair, cell apoptosis, and cell proliferation and differentiation [14,15]. Autophagy has a dual effect. Previous studies have shown that autophagy is necessary for airway fibrosis as a reserve of energy to provide cellular resources required for extracellular matrix protein biosynthesis. However, autophagy can also inhibit airway fibrosis by degrading extracellular matrix (ECM) proteins and their precursors [16]. There is increasing evidence that autophagy plays a critical role in important cellular processes, including tissue remodeling, and is an effective process for the extensive degradation and circulation of cytoplasmic components to maintain basal cellular homeostasis and healthy organelle populations within cells [17]. Tissue remodeling and fibrosis are common features of many airway diseases and cannot be prevented or reversed by current therapies. Autophagy has been regarded as an important target for the treatment of some diseases. However, there are few studies on autophagy and RA-ILD [12,18,19,20,21].

Under various conditions, autophagy occurs and forms autophagosomes, which combine with lysosomes and form autophagolysosomes. Then, autophagolysosomes begin to degrade, and some junction proteins, such as P62, link the aggregation proteins and damaged organelles with autophagic degradation. When these junction proteins are abnormal, they can lead to disruption of cellular signaling and metabolic pathway homeostasis, with important implications for countless metabolic diseases [22], neurodegenerative diseases, and so on [23].

However, it has been found in the existing studies of RA that resveratrol (Res) can improve the progression of the disease by increasing the autophagic flux, and the use of resveratrol can reduce the severity of arthritis [24]. Resveratrol (Res) is the most famous polyphenol class of Astragalus, found in grapes, mulberries, peanuts, rhubarb, and several other plants, which may play a beneficial role in the prevention and promotion of chronic diseases related to inflammation, such as diabetes, obesity, cardiovascular disease, neurodegenerative diseases, and cancer [25,26,27]. In previous studies, it was found that resveratrol, as a natural regulator of autophagy, has minimal side effects and well applied prospects and has been proven to play an important regulatory role in various aspects, such as aging [28] and cancer [29].

Therefore, we seem to be able to link resveratrol with autophagy and RA-ILD. In this study, we investigated the protective mechanism of resveratrol on RA-ILD and emphasized the regulation of autophagy levels. We first established the experimental animal model induced by collagen and adjuvant combined with bleomycin (including the pre-experimental model) to determine the change in autophagy level, and then observed the change of autophagy after resveratrol (experimental animal) treatment, to explore the relationship between autophagy and fibrosis. Finally, by simulating the disease environment in human embryonic lung fibroblasts (MRC-5 cell), it has been proven that resveratrol improved the level of fibrosis by regulating autophagy.

## 2. Results

### 2.1. An Animal Model of Rheumatoid Arthritis-Related Interstitial Lung (RA-ILD) Was Established

An animal model of RA-ILD was established (Figure 1A). To verify the success of model establishment, we compared the lung tissue weights of the different groups (Figure 1B). The results showed that the wet lung weight of the model group increased, and the differences in the lungs of mice in the model group were preliminarily determined (*p* < 0.05). To judge the establishment of the model more intuitively and accurately, we performed H&E staining (Figure 1C) and Masson’s staining (Figure 1D) on the lung tissues of mice. According to the lung tissue performance of mice in the CIA+BLM group, the lung tissue structure of mice in the experimental group was significantly changed, with local pleural thickening and alveolar wall thickening. The structure was generally disorganized with inflammatory cell infiltration. At the same time, there were some blue changes in the interstitial tissue observed by fiber staining, which represent fibrosis, and especially cord-like blue fibers exist. The CIA+BLM model was mainly dominated by extensive pulmonary interstitial fibrosis changes. Then, we further measured the hydroxyproline content (Figure 1E), which was significantly higher in the CIA+BLM group (*p* < 0.01). Subsequently, genes in lung tissue were measured (Figure 1F), and the mRNA levels of inflammatory factor (IL-1β), fibrotic markers (COL1A1, COL3A1, FN1), and TGF-β in the CIA+BLM group were significantly higher than those in the control group.

### 2.2. Related Proteins in the Lungs of the CIA+BLM Group Were Changed 

Considering that mRNA levels were significantly changed, subsequently, the levels of autophagy-related proteins in lung tissues were detected (Figure 1G). The results showed that autophagy-related proteins in lung tissues were upregulated and the level of the reactive oxygen species-associated protein HIF-1a was also significantly upregulated. Especially, the ratio of the key autophagy protein LC3 type II to type I (LC3 II/I) was significantly increased (*p* < 0.01).

### 2.3. Resveratrol Improved the Progression of CIA+BLM, but Had Less Effect on Lung Inflammation

In view of the successful establishment of animal models, we performed a resveratrol intervention to investigate whether resveratrol contributes to the progression of RA-ILD. Combining the results of Figure 1 to identify the stage of fibrosis progression [30,31], referring to the previous literature on resveratrol [26,32], the progression of CIA+BLM was interfered with by the administration of resveratrol (Figure 2A) to determine the effect of resveratrol on pulmonary fibrosis. We performed H&E staining on joints (Figure 2B), confirming the therapeutic effect of resveratrol on RA. The CIA+BLM group in the experimental group showed joint synovial cell proliferation and infiltration and obvious inflammation, with a high pathological score of arthritis (Figure 2C, *p* < 0.0001). After resveratrol treatment, the pathological performance of arthritis was weakened (*p* < 0.05). At the same time, the images of lung tissues were observed (Figure 1D), and the results showed that CIA+BLM lung tissues were pale and larger in volume and were slightly smaller after resveratrol treatment. Later, the wet weight of lung tissues and the ratio of wet weight to dry weight (Figure 2E,F) were statistically analyzed. The results show that, except for the results corresponding to Figure 1B, there was no significant change in the wet weight of lungs after resveratrol treatment, and the wet–dry weight ratio had no difference with the control group. The appearance of lung tissue was not acute edema but some chronic changes. Lungs were observed by H&E staining (Figure 2G) more intuitively, and the structural changes of lung tissue after resveratrol treatment were not as obvious as those in the experimental group, and inflammatory cell infiltration was only rarely distributed near the pleura. Furthermore, the inflammation score (Figure 2H) was determined in the tissues, and it was found that inflammatory changes were not prominent after resveratrol treatment, which means resveratrol did not increase the level of inflammation in the lungs. This was also confirmed by subsequent gene level detection of the inflammatory cytokine IL-1β (Figure 2I).

### 2.4. Resveratrol Attenuated Fibrosis in Lung Tissue

Since the results show that resveratrol may have an effect on lung chronic changes in CIA+BLM mice, we focused on the observation and analysis of fibrosis, the main pathological manifestation of RA-ILD. We continued histological staining of lung tissues. Masson’s staining (Figure 3A), which was specific for fibrosis, showed that after resveratrol treatment, the blue collagen fibers in lung tissues were reduced, which was different from the obvious rod-like fibers that existed in the lung interstitial in the CIA+BLM group, and the incidence of extensive interstitial fibrosis was low. To explain the results more clearly, we conducted statistics on the area of collagen staining in lung tissues (Figure 3B), and it was obvious that the overall fibrotic area of lung tissues in the CB+Res group was reduced (*p* < 0.05). At the same time, the pulmonary tissue fibrosis score was determined (Figure 3C), and the results were consistent with the previous results. The fibrosis score of the CB+Res group decreased (*p* < 0.05). In addition to these findings, lung tissue hydroxyproline content was also measured (Figure 3D), showing that the content of hydroxyproline in the CIA+BLM group increased (*p* < 0.0001), and it decreased (*p* < 0.01) after resveratrol treatment, which further confirmed the change in chronic fibrosis in lung tissue, and the fibrosis of lung tissue was improved after resveratrol treatment. Therefore, a CIA+BLM animal model was successfully established, and it was preliminarily found that resveratrol treatment improved pulmonary fibrosis. Furthermore, the expression levels of FN-1 (Figure 3E), COL13A1 (Figure 3F), and protein (Figure 3G), representing fibrosis in lung tissues, were also detected. The results all confirmed the previous results: the expression levels of the fibrosis genes FN-1 and COL3A1 were significantly decreased after resveratrol treatment (*p* < 0.01), and the protein expression level of collagen I was also significantly downregulated (*p* < 0.01). These results indicate that both tissue expression and gene and protein expression levels confirm the beneficial effect of resveratrol on fibrosis.

### 2.5. Resveratrol Regulated Autophagy in CIA+BLM and Affected Oxidative Stress-Related Proteins

Considering resveratrol’s effect on major pathological changes in the lungs of CIA+BLM, we considered autophagy, which plays a critical role in maintaining metabolic homeostasis in lung tissues and the development of chronic respiratory diseases. The expression level of the key autophagy gene the LC3B in lung tissues was detected (Figure 4A), which showed that the mRNA level of LC3B decreased after resveratrol treatment (*p* < 0.01). Correlation analysis of related genes (Figure 4B,C) showed that the LC3B gene level was positively correlated with COL3A1 (*p* < 0.0001 and r = 0.9258) and FN1 (*p* < 0.01 and r = 0.6389). That is, with the increase in the fibrosis gene level, the expression level of the key autophagy gene LC3B increased. LC3B represents the total of LC3 I and LC3 II. However, the total expression level of the LC3B gene could not reveal the transformation process from type I to type II but could also not judge the change in the autophagy process. Therefore, protein expression levels were further detected (Figure 4D,E). The results showed that the expression of oxidative stress-related proteins was increased in the CB+Res group, while the transformation of LC3 I to LC3 II was higher than the control (*p* < 0.01), but not increased compared to CB. To explain this expression, we detected other key autophagy genes, P62 (Figure 4F) and BNIP3 (Figure 4G). The expression level of P62, an important autophagy receptor (also known as SQSTM1 protein), was significantly decreased after resveratrol treatment (*p* < 0.05), while the expression of BNIP3 was not significantly changed. It was also found that P62 was positively correlated with the expression of the fibrosis-related gene FN1 (Figure 4H, *p* < 0.001 and r = 0.7479). The detection of protein expression (Figure 4I,J) showed that the autophagy-related proteins BNIP3, BECN1, and BCL2 did not change significantly in the CB+Res group. However, when CIA+BLM expression was significantly increased, the level of P62 decreased after resveratrol treatment (*p* < 0.05), while the level of phosphorylated P62 increased. This indicates that the autophagy level of CB+Res changed significantly, and it may be related to the protein degradation process, which may be responsible for resveratrol’s ameliorative effect on pulmonary fibrosis.

### 2.6. Resveratrol Reversed Disruption of Autophagosome–Lysosome Fusion In Vitro to Improve Fibrosis

It was confirmed that resveratrol did regulate the progression of autophagy and fibrosis and had a greater relationship with the later stage of autophagy. However, whether resveratrol improves fibrosis by regulating the late autophagy process needs to be further verified in vivo. Instead of simulating IPF disease models with TGF-β alone [33], we treated MRC-5 (human embryonic lung fibroblasts) with IL-1β and TGF-β (Figure 4A), which was also consistent with previous mRNA expression levels (Figure 1F). After treatment, the protein levels of fibrotic Collagen I and autophagy marker protein LC3 II/I in the model group were significantly increased (*p* < 0.5), and the important autophagy receptor P62 was abnormally accumulated (*p* < 0.5). However, after intervention with different concentrations of resveratrol (Figure 5A), the expression level of fibrotic protein gradually decreased, the level of LC3 II/I did not increase or even decreased, and the protein level of P62 was significantly reversed. That is, with the increase of Res concentration, the fibrosis level of the model group gradually decreased, and the degradation of P62 was more unobstructed. However, whether resveratrol targets autophagy to improve fibrosis remains to be verified. The cells in the model group were treated with different concentrations of the classical late autophagy inhibitor CQ (Figure 5B). The protein expression level showed that with the increase of CQ concentration, resveratrol reversed the fibrosis level in the model group. When the concentration of CQ was 40 μM, the reverse effect was the most obvious, the fibrotic protein Collagen I was significantly increased (*p* < 0.001), there was abnormal accumulation of P62 (*p* < 0.05), and the LC3 II/I level was also increased (*p* < 0.01). This indicates that the mechanism of Res improving fibrosis by regulating autophagy is established. To further explain the mechanism of resveratrol action, we added two inhibitors, BafA1 and MG132, for comparison of several possible mechanisms affecting degradation in the late stage of autophagy (Figure 5C). The results showed that although BafA1 and MG132 also affected the degradation process of P62, the level of P62 recovered after treatment (*p* < 0.5), but it could not completely reverse the inhibitory effect of resveratrol on the level of fibrosis in the model group. In contrast, the result after CQ treatment further proved that resveratrol improved the process of fibrosis by regulating autophagy. However, the specific mechanism may be related to the fusion process between lysosomes and autophagosomes (the effect of CQ). Therefore, for further confirmation, mCherry-GFP-LC3B plasmid was transferred into MRC-5 cells (Figure 5D), and the fusion process of autophagosome and lysosome was judged by observing the change of fluorescence (Figure 5E). The results suggested that autophagic flow increased significantly after resveratrol treatment and that resveratrol reversed the disruption of autophagosome–lysosome fusion to ameliorate fibrosis in vitro. Therefore, based on the above experimental results, we summarized the corresponding pattern diagrams so as to show more clearly the content of the article study (Figure 6).

## 3. Methods

### 3.1. Animal Experiments

#### 3.1.1. Animal Experiments

Eight-week-old male C57BL/6 mice (body weight: 20–25 g) were randomly divided into 5–8 mice in 6 groups: (I) Normal, (II) Ctrl (Control), (III) BLM (bleomycin, 3.5 mg/kg), (IV) CIA (collagen-induced arthritis models), (V) CIA+BLM (CIA- and bleomycin-treated models), and (VI) CB+Res (CIA+BLM- and resveratrol-treated models, 10 mg/kg).

Male C57BL/6 mice were purchased from Jiangsu Ji Cui Biotechnology Co, Ltd. (Nanjing, China), and adapted to laboratory conditions for at least 1 week at Anhui Medical University before the experiment: conventional breeding, with independent ventilation to cage box (IVC) individual ventilated cages in the animal housing, and fed with a standard animal diet and water.

We used the previous modeling method of chicken type II collagen and complete Freund’s adjuvant combined with bleomycin [30,34]. CIA mice were induced by both CFA and Col II. Then, 2 g/L of chicken Col II (Chondrex, Woodinville, WA, USA) was dissolved in 0.05 mol/L of glacial acetic acid, fully stirred, and then stored overnight at 4 °C. CFA (Chondrex, Woodinville, WA, USA) was then mixed with an equal volume of dissolved Col II and fully emulsified in the ice bath of the homogenizer. Mice treated with BLM were anesthetized by intraperitoneal injection of 5% chloral hydrate on the 25th day, and tissues were separated layer-by-layer to expose the trachea. The micro-syringe absorbed BLM solution at 3.5 mg/kg (BLM, Life Technologies, Carlsbad, CA, USA, the finished product concentration of 100 mg/mL, diluted with normal saline to a final concentration of 1.5 mg/mL) and was slowly dropped into the trachea. The mice were rotated upright while dropping so that the liquid could be fully and evenly distributed in the lungs, and the breathing of the mice could be observed to prevent suffocation. 

CIA+BLM and CB+Res groups were injected with 100 μL of emulsion on day 0 and day 21. Then, BLM was administered on day 25. After normal feeding for 10 days, animals in the CB+Res group were administered resveratrol (Aladdin, Shanghai, China) dissolved in 0.5% sodium carboxymethyl cellulose (carrier) by gavage once a day, with a final concentration of 1 mg/mL, and each animal was given 0.1 mL/10 g for a total of 10 days. 

The control group was injected with the same volume of normal saline through the trachea and tail root as a negative control. The normal group was not treated. The CIA+BLM group was used to construct an RA-ILD animal model [30], and the resveratrol dose was based on previous reports [32,35]. Resveratrol treatment referred to CIA+BLM pathological changes at different times [30,31].

All animals were sacrificed 24 h after the final resveratrol treatment. The lung tissue was taken for photographing and weighed for retention, and the size of soybean grains was cut off for dry and wet weight measurements. The knee and one lung tissue were collected for histological analysis, and the remaining lung tissues were immediately frozen in liquid nitrogen for subsequent biological analysis.

#### 3.1.2. Lung Hydroxyproline Content

Lung hydroxyproline content was determined by standard spectrophotometry according to the kit instructions (Solarbio, Beijing, China).

#### 3.1.3. Lung Dry Weight and Wet Weight

The lung tissues the size of a soybean grain were dried overnight in a 56 °C oven. The dry weight was weighed and compared with the wet weight before drying to obtain the wet-to-dry weight ratio of lung tissue.

#### 3.1.4. Hematoxylin–Eosin (H&E) and Masson’s Staining

After fixation with 4% neutral formaldehyde, the tissue specimens of the knee joint were placed in EDTA decalcification solution (Boster, Wuhan, China) in a normal temperature shaker for 2 weeks, and then embedded in conventional dehydrated paraffin. The thickness of the continuous sections was approximately 4.5 um, and the tissue was attached to an anti-decalcification slide and then placed in a 60 °C incubator for baking. The slices were dewaxed and washed with gradient alcohol and distilled water. Hematoxylin was performed for 5 min, 2 h after flushing with tap water, to turn the sample blue. After dyeing with 5% eosin for 40 s, the sections were quickly dehydrated and cleared. Finally, they were sealed with neutral gum for H&E staining. We used a histological score system to evaluate individual joints and assess arthritis severity [36].

The lung tissue specimens were fixed, dehydrated, paraffin-embedded, and cross-sectioned successively to a thickness of approximately 3.5 um. One part was stained with H&E staining, and the joint was also stained. A Masson staining kit (Solarbio, Beijing, China) was used for some of the tissue sections. The cells were stained with iron hematoxylin to stain the nucleus for 8 min at room temperature and then differentiated into the blue. Aniline blue staining was performed for 1.5 min, with weak acid washing and dehydration by sealing tablets. Finally, the slices were observed under a microscope (Nikon Eclipse 80I) and analyzed with Image Pro 6 (Media Contronetics Inc., Bethesda, MD, USA). Histological changes in randomly selected histological fields were assessed at 400× magnification [37].

The lung pathology inflammation score was divided into grades 0–4 according to previous statistical methods [38]. The Ashcroft scoring method was used to score the fibrosis level of each group in Masson-stained sections on a scale of 0 to 8 [39]. Then, the section staining map was imported into Image Pro 6 (Media Contronetics Inc. Bethesda, MD, USA), and the fibrotic area was counted [37].

#### 3.1.5. Western Blotting 

Total proteins were extracted from model mouse lung tissue and cells using a prepared SDS lysis buffer for Western blotting analysis. Then, 60 ug of protein was isolated using 12% SDS-polyacrylamide gel and imprinted onto polyvinylidene fluoride membrane (PVDF, Merck Millipore, Germany). Then, the membrane was sealed in 5% skim milk for 1.5 h. β-actin (Abcam, # ab8226), Collagen I (Arigo Biolaboratories, #ARG21965), HIF-1a (Proteintech, #20960-1-AP), LC3A/B (Cell signaling technology, #12741), P62 (Cell signaling technology, #16177S), Phospho-SQSTM1/p62(Ser349) (CST, #E7M1A), Bcl-2 (Abcam, #ab182858), BNIP3 (Santa Cruz Biotechnology, #sc-56167), BNIP3L (Proteintech, #12986-1-AP), BECN1 (Boster, #PB0014), and SOD2 (Boster, #BA4566) were used. The PVDF membranes were subjected to incubation with antibodies above at 4 °C overnight, then HRP-labeled secondary antibody (ZSGB-BIO, Beijing, China) was added and incubated for 1 h, and the immune reaction was detected by the ECL luminescence system.

#### 3.1.6. Quantitative RT-PCR

Total RNA was extracted from mouse lung tissues using TRIzol reagent (Thermo Fisher Scientific, Waltham, MA, USA). In short, TRIzol was added to a dry powder ground on lung tissue and left to stand at room temperature for 5 min. Then, 200 μL of chloroform was added to each sample, followed by a severe upside-down vortex. After standing for 5 min, the samples were centrifuged at 12,000 rpm and 4 °C for 15 min. The samples were divided into 3 layers. The aqueous phase was reserved with 400 μL of isopropyl alcohol, left for 10 min, and centrifuged at 12,000 rpm at 4 °C for 10 min. The supernatant was removed, washed with 70% ethanol, centrifuged, further dried, and then suspended in DEPC water. Finally, the concentration and purity of the extracted RNA were determined by ultraviolet spectrophotometry (One Drop, New York, NY, USA). According to the instructions of the cDNA Reverse Transcription Kit (Beyotime, Shanghai, China), primers were used to reverse-transcribe total RNA. Quantitative PCR was performed using SYBR Green (Beyotime, Shanghai, China) in a Roche LightCycler^®^ 480 real-time PCR system. After the reaction was completed, the period threshold (Ct) was determined, and the relative RNA levels of each sample were calculated using the 2-ΔΔCt method and normalized to the GAPDH level. The gene-specific primers used in this study are listed in Table 1.

### 3.2. Cell Experiments

#### 3.2.1. Cell Culture and Reagents

The human fetal lung fibroblast cell line (MRC-5) was purchased from American Type Culture Collection (ATCC, Rockville, MD, USA) and cells were cultured in Dulbecco’s modified Eagle’s medium (DMEM, Life Technologies/Gibco, Grand Island, NY, USA) supplemented with 10 % (*v/v*) fetal bovine serum (Gibco) and 1% antibiotic-antimycotic solution (containing 100 U/mL penicillin, 100 µg/mL streptomycin, and 0.25 µg amphotericin B; Sigma-Aldrich, St. Louis, MO, USA). Cells were maintained in a humidified air atmosphere with 5% CO_2_ at 37 °C. 

To induce the cell model of RA-ILD along with the animal model, recombinant TGF-β1 and IL-1β were added to the cell medium for 24 h, purchased from MedChemExpress (Monmouth Junction, NJ, USA). Chloroquine (CQ) and Bafilomycin A1 (BafA1) were purchased from MedChemExpress (Monmouth Junction, NJ, USA). MG132 was purchased from Selleckchem (Houston, TX, USA).

#### 3.2.2. Cell Transfection

Cells were transiently transfected with mCherry-GFP-LC3B constructs using EZ Cell Transfection Reagent (Shanghai Life-iLab Biotech, AC04L091), as described by the manufacturer. Briefly, 50,000 cells were transfected with 0.5 µg of mCherry-GFP-LC3B plasmid and cultured in growth media with an exchange of medium every 24 h. Two days after transfection, cells were used for treatments and confocal imaging. After the treatment of cell models for 48 h, the numbers of mCherry and GFP fluorescence points were directly observed under the laser confocal microscope (Leica SP8, Wetzlar, Germany). 

### 3.3. Statistical Analysis

All results are presented as mean ± standard error of the mean (SEM). The overall significance of the results was examined by one-way ANOVA using GraphPad Prism 5 (GraphPad Software, La Jolla, CA, USA). To determine differences between compared groups, *p* < 0.05 was considered statistically significant.

## 4. Discussion

RA-ILD is a complication of RA with high morbidity and mortality, and its treatment has been focused on fibrosis [6,40]. Pulmonary fibrosis is a process of malfunctioning tissue repair. Autophagy, which plays an important role in tissue repair, has been proven to play a role in other respiratory diseases and has even been regarded as an important therapeutic target [16]. In the existing studies on RA-ILD, there have been few studies on the mechanism of this disease, mainly due to the difficulty of animal model construction. Several animal models of rheumatoid arthritis have been used to replicate articular pathology [5], but few animal models have shown pulmonary pathology as an extra-articular manifestation of the disease, mostly in the absence of fibrosis and UIP features, and their application in experimental studies has been limited [5,41]. Different from the model of pulmonary fibrosis induced by BLM alone or CIA alone, the animal model constructed by CIA combined with BLM showed significant interstitial fibrosis and cellular changes, accompanied by dilated air gaps and thickened interstitial walls, most notably the proximity of collagen fibers to the pleura. These behaviors were consistent with typical UIP characteristics [3,4,6,41]. Combined with the manifestations of arthritis and pulmonary fibrosis in the experiment, this fully demonstrates that the CIA+BLM model can be used to better study the development of RA-ILD disease and provides a possibility for the study of its intervention in the development of fibrosis. By detecting the protein and gene levels of the model mice, in the experiment, we found that the autophagy level of the model mice changed significantly. Autophagy changes are associated with the occurrence of disease.

In addition, it could be found that the key autophagy proteins BNIP3, BECN1, and BCL2 changed significantly, and the oxidative stress-related protein levels also changed, pre-experiment and post-experiment. Tissue injury usually results in a temporary loss of normal vascular perfusion and changes in the expression of hypoxia-induced genes [42]. Current studies suggest that mitochondrial autophagy can prevent the accumulation of damaged mitochondria and increase reactive oxygen species homeostasis levels, leading to oxidative stress and cell death [43]. Meanwhile, most of the factors that promote pulmonary fibrosis, such as oxidative stress, endoplasmic reticulum stress, and hypoxia, can induce autophagy [43]. 

Studies have shown that BNIP3 and BNIP3L can activate autophagy by ectopic expression under nonmonic conditions, and the hypoxia-induced atypical BH3 domain of BNIP3/BNIP3L is designed to induce autophagy by disrupting the BCl2–BECN1 complex [44]. BECN1, the central component of the PI3K initiation complex during the initiation of autophagy, interacts with BCL2 to regulate the size and number of autophagosomes [45]. Autophagy and oxidative stress-related protein levels are closely related in RA-ILD. The occurrence of RA, an autoimmune disease, leads to pulmonary inflammation and tissue repair disorder, extracellular matrix deposition, and tissue fibrosis. These tissue damage factors result in a transient loss of corresponding normal vascular perfusion, resulting in an anoxic microenvironment. Under hypoxic conditions, the corresponding hypoxia-inducible factors such as HIF-1a protein increase, and the oxidative stress level increases. A growing amount of evidence in recent years argues for oxidative stress acting as the converging point of the stimuli (damage factors) that cause autophagy, with reactive oxygen species (ROS) being among the main intracellular signal transducers sustaining autophagy [46].

Therefore, ROS-related proteins acted as maintainers during the autophagy in CIA+BLM. After stimulation by the tissue damage factors under sustained ROS, BNIP3 is elevated, so that the BCl2–BECN1 complex is competitively destroyed, while BECN1 is released, and autophagy occurs. That is, when disease stimulates, autophagy occurs, attempting to prevent the occurrence of autoimmune scenarios by clearing pathogens in cells and maintaining the homeostasis of immune cells through autophagy. Therefore, abnormal levels of autophagy were found in lung tissue.

However, resveratrol, which has been shown to be effective in rheumatoid arthritis, was used to interfere with the autophagy process occurring in lung tissue. The results show that resveratrol has effects on fibrosis of lung tissue, and lung inflammation, such as methotrexate, did not aggravate the effect. These findings were also consistent with the treatment direction of RA-ILD [6]. Not only that, in lung inflammation and fibrosis, fibrosis-based treatment is effective. From the experimental results, resveratrol significantly improved fibrosis at the gene, protein, and tissue levels, and prevented the progression of fibrosis.

After the clear effect of resveratrol on the progression of fibrosis, its mechanism has become the focus of research. Focusing on the level of autophagy in CIA-BLM-treated mice, we noted that the level of LC3, a key marker protein of autophagy, was increased, and LC3 was gradually transformed from a cytosolic form (LC3-I) to a membrane-bound lipidosis form (LC3-II), indicating that autophagy was still activated in the early stage and autophagosome formation was increased. Interestingly, P62 (SQSTM1), an important autophagy receptor, was abnormally elevated in the experimental group. Resveratrol reversed this abnormal rise. During autophagy, P62 binds to ubiquitinated proteins and then forms a complex with LC3-II protein located on the autophagic micro-membrane, which degrades into autophagic lysozyme [22]. Phosphorylation of S409 at P62 regulates the dimer interface of the UBA domain of P62, enhancing the affinity between P62 and ubiquitin, while a lack of phosphorylation damages the recycling of autophagy proteins and the degradation of ubiquitinated proteins or Poly Q-Expansion proteins. However, selective autophagy can be triggered by ULK1-dependent phosphorylation of P62 to regulate the clearance of ubiquitinated proteins or aggregative precursor disease proteins [22]. The results showed that although autophagosomes were increased, the degradation process of P62 was inhibited. That is, the downstream of autophagy is inhibited, while autophagy flow is blocked, and autophagy flux is still small, which hinders the purposeful degradation process. The big question is whether resveratrol releases this blocking state. We found that resveratrol restored the blocked autophagy flow, it did not significantly change LC3II/I, but significantly promoted the degradation of abnormally accumulated P62, regulated autophagy, and improved the progression of pulmonary fibrosis. LC3 II in the CB group will be blocked due to autophagy, resulting in the deposition of LC3 protein, which leads to the increase of LC3 II/I. However, after resveratrol treatment, autophagy flow was smooth, and the conversion of LC3 I to II was increased. At the same time, resveratrol also cleared the deposited LC3 and other proteins, so LC3 II/I did not increase. In the cell model, LC3II/I even decreased with the increase of the resveratrol concentration, which was the result of autophagy overactivation after resveratrol. The decrease in LC3B (Figure 4A) can also be explained here. Resveratrol did not significantly increase the signaling of BECN1, an autophagy regulator, suggesting that resveratrol may also support atypical (Beclin-1 independent) autophagy pathways during CIA+BLM.

After understanding the changes in the autophagy process of the disease, to better verify the regulation of autophagy participating in fibrosis improvement, we continued to study the relationships. We found that abnormal elevation of the P62 gene and the abnormal elevation of LC3B caused by LC3B-II deposition were correlated with the changes in fibrosis gene levels. Reducing P62 levels after resveratrol treatment may improve the expression of fibrosis genes.

However, in addition to demonstrating that resveratrol has regulatory effects on autophagy and fibrosis in animal tissues, to better verify the specific mechanism of resveratrol and whether autophagy improves fibrosis, we established a cell model consistent with the animal model. In previous studies, only TGF-β-induced cells were used to establish a pulmonary fibrosis model or specific fibrosis disease model, without a specific RA-ILD cell model. To differentiate RA-ILD from other common pulmonary fibrosis and to be in line with the highly mixed background of inflammation and fibrosis in RA-ILD disease, we combined the abnormal elevation of IL-1β and TGF-β in an animal model, and the combined induction of human embryonic lung cells can better simulate the disease environment of RA-ILD in a human environment. The results also proved that the late degradation process of autophagy was also blocked in the combined treatment cell model. After resveratrol intervention, fibrosis improved, and abnormally accumulated proteins were degraded. This undoubtedly provides a good condition for further research.

There are two key degradation systems for protein degradation in nature, and whether the proteasome or autophagolysosome degradation pathway is blocked in the degradation process is mainly determined by adding corresponding inhibitors. After treatment with autophagy inhibitors, the autophagy process was blocked, P62 accumulated again, and the ameliorative effect of resveratrol on fibrosis was reversed. The effect of MG132 and BafA1 was not as good as that of CQ. Therefore, we better verified that the ameliorating effect of resveratrol on fibrosis in the context of disease is achieved by regulating the late degradation process of autophagy.

Then, combined with previous functional studies of CQ [47], we hypothesized that the effect of resveratrol may be related to the fusion process between lysosomes and autophagosomes, which may increase autophagic flux by promoting autophagosome–lysosome fusion. We also distinguished the mCherry-GFP-LC3B autophagosomes (GFP and Cherry-positive LC3 spots, thus yellow) from the more acidic autolysosomes (GFP-negative and Cherry-positive LC3 spots, thus red). Under the condition of non-autophagy, mCherry-GFP-LC3B existed in the cytoplasm in the form of yellow fluorescence under the fluorescence microscope. After autophagy occurred, the fluorescence was concentrated on the autophagosome membrane in the form of yellow spots. After the fusion of autophagosomes and lysosomes, the autophagosomes were partially quenched by GFP fluorescence and appeared as red spots. Therefore, there were fewer red spots in the RA-ILD model group, while the red spots increased and the green spots decreased after resveratrol treatment, which was due to the obstruction of the fusion process between autophagosome and autophagolysosome in the model group, and the later stage of autophagy was blocked. However, CQ can reverse the effect of resveratrol, and it was further proven that resveratrol plays a role in regulating the late degradation process of autophagy by promoting the fusion of autophagosomes and lysosomes.

The results suggest that the occurrence of fibrosis may be related to the abnormality of key autophagy proteins to some extent. The experimental results suggest that in RA-ILD, the occurrence of fibrosis leads to the activation of autophagy, and the body attempts to clear pathogens by autophagy. Autophagosomes are formed after activation of the pro-autophagy process. However, the further degradation process is blocked in the later stage, and the autophagy flux is inhibited, resulting in the final deposition of pathogens that should be removed, which cannot be removed but may exacerbate the disease progression. Thus, although there may be early activation of autophagy during the disease process, autophagy fails to play an adequate role due to late inhibition. However, after the use of resveratrol, autophagy circulation can be free, autophagy is activated as a whole, tissue repair ability is increased, and sediment clearance is sufficient to achieve the therapeutic effect. Similarly, resveratrol enhances autophagic flux by decreasing P62 expression, showing cardiac and neuroprotective effects [48]. In addition, we noted that P62(SQSTM1) interacts with KEAP1, a cytoplasmic inhibitor of NRF2, which is a key transcription factor involved in the cellular oxidative stress response. P62(SQSTM1) is associated with oxidative stress [49]. This also confirmed that oxidative stress-related protein levels were also abnormally elevated in the presence of abnormally elevated P62 levels.

Resveratrol, a well-known polyphenol, has been shown to affect the progression of RA by different mechanisms [50,51,52,53]. Our results are consistent with previous studies on the effect of resveratrol on arthritis, and its therapeutic effect on arthritis can be proven. Several recent studies have suggested that drug-enhanced autophagy flux may have disease-improving activity for RA [54,55]. The enhancement of the autophagy pathway is associated with RA [56,57], and resveratrol reduces the severity of experimental rheumatoid arthritis by activating autophagy. The amount of resveratrol ester used in the experiment (10 mg/kg/d) and data from clinical trials indicate that resveratrol is safe and well-tolerated at daily doses between 20 mg and 2 g [58]. Resveratrol has a preventive effect on bleomycin-induced pulmonary fibrosis, but there have been few studies on the treatment of fibrosis [59]. From the experimental results, it is obvious that resveratrol can improve histopathology, and the changes in the protein and gene levels of fibrosis: the therapeutic effect of resveratrol is worth further study. In the experiment, resveratrol’s therapeutic effect is worth affirming. 

There is no doubt that autophagy is a hotspot in biomedical research. Its role in aging and in the development of human pathology includes many rheumatic diseases, including rheumatoid arthritis (RA) [60]. In the treatment of rheumatoid arthritis, commonly prescribed drugs, such as methotrexate, leflunomide (LEF), and antitumor necrosis factor-alpha agents, are beneficial to the joints, which are associated with the occurrence and acceleration of existing ILD in vitro [7]. While we look for new directions in drugs for fibrosis, we can focus on some other drugs that are beneficial for arthritis. As well as treating arthritis, we can also prevent or treat complications such as interstitial lung disease. The mechanism by which resveratrol improves fibrosis, proved by the results of this study, is similar to that of its treatment of arthritis, which will provide an important and promising idea for the treatment of RA and RA-ILD.

However, it should be noted that the treatment effect of resveratrol was inhomogeneous, and the degree of fibrosis improvement was different, which might be related to the individual treatment differences of resveratrol. The deeper mechanism of its therapeutic effect might be clearly understood if there is a gradient setting of resveratrol’s dosage. Future studies should further explore the effects of different drug doses of resveratrol on treatment and a deeper mechanism by which resveratrol regulates autophagy, to open a new window for the treatment of RA-ILD.

## Figures and Tables

**Figure 1 molecules-27-08475-f001:**
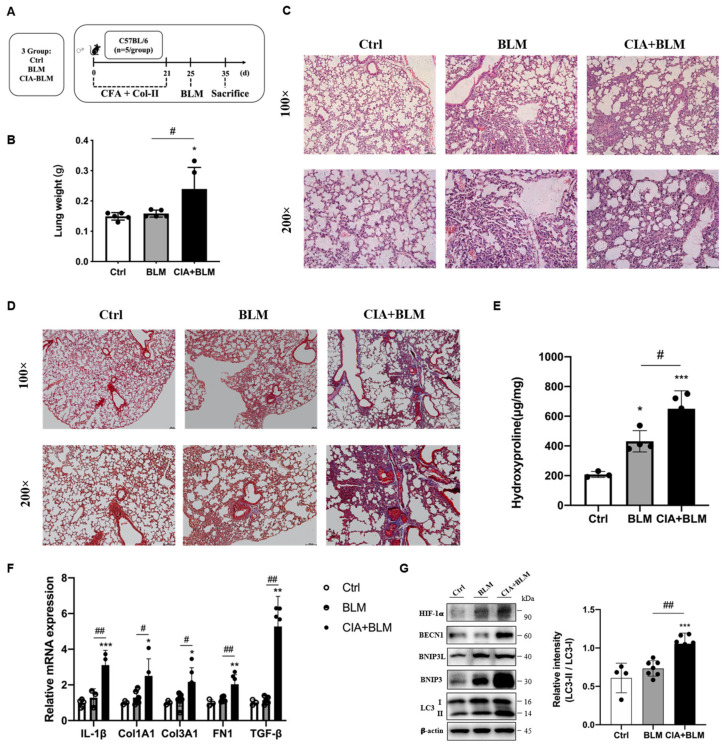
Animal models of RA-ILD were constructed, and the expression of autophagy-related proteins was changed. (**A**) CFA and chicken collagen type II were subcutaneously injected into the tail root of C57BL/6 mice on day 0 and 21 and then treated with bleomycin on day 25. On day 35, the mice were sacrificed. (**B**) The lung weight was measured. (**C**,**D**) Representative H&E and Masson’s staining images of lungs from the three groups. Scale bars = 50 μm, shown at 100× (top panels) and 200× (bottom panels). (**E**) The levels of hydroxyproline in the lung were measured. (**F**) The mRNA levels of IL-1β, Col1A1, Col3A1, FN-1, and TGF-β were quantified. (**G**) The levels of HIF-1a, BECN1, BNIP3L, BNIP3, LC3 I, and LC3 II in the lungs were detected using immunoblotting. The ratios of LC3 II/I were quantified as described above. Values represent the mean ± SEM (n = 3–5). Compared with the Ctrl group, * *p* < 0.05, ** *p* < 0.01, and *** *p* < 0.001; compared with the model group (BLM), # *p* < 0.05 and ## *p* < 0.01.

**Figure 2 molecules-27-08475-f002:**
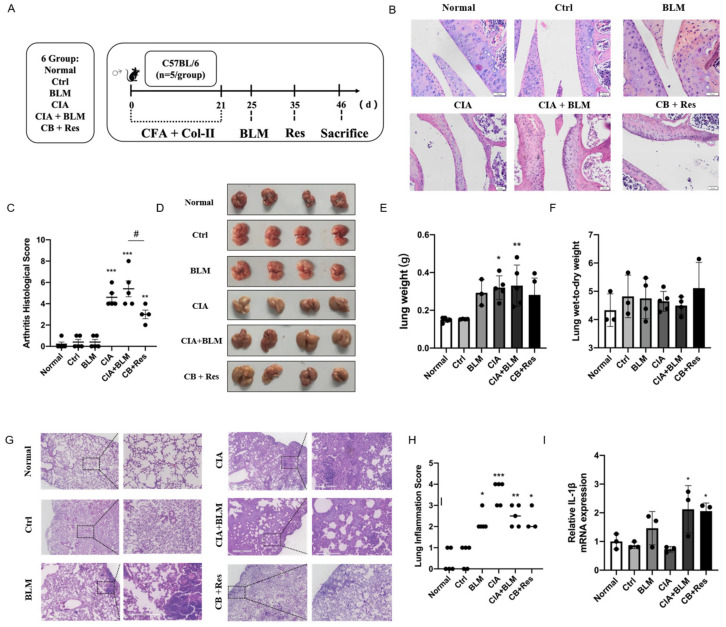
Resveratrol attenuated the inflammation of arthritis, and it did not aggravate lung inflammation in CIA+BLM mice. (**A**) CFA and chicken collagen type II were subcutaneously injected into the tail root of C57BL/6 mice on day 0 and 21, then treated with bleomycin on day 25, and resveratrol was taken orally on day 35, which lasted for 12 days. Then, the mice were sacrificed on day 46. (**B**) Representative H&E staining image of arthritis from each group of the study. Scale bars = 50 μm, 40× (**C**) The arthritis pathology score was estimated. (**D**) The lungs of each group are shown. (**E**,**F**) The lung wet weight and ratio of wet weight to dry weight were measured. (**G**) Representative image of pulmonary H&E staining shown at 40× (left panels) and 200× (right panels) magnification. Scale bar, 50 μm. (**H**) Assessment of inflammation in every group. (**I**) The mRNA levels of IL-1β were quantified. Values represent the mean ± SEM (n = 3–5). Compared with the normal group, * *p* < 0.05, ** *p* < 0.01, and *** *p* < 0.001; compared with the model group (CIA+BLM), # *p* < 0.05.

**Figure 3 molecules-27-08475-f003:**
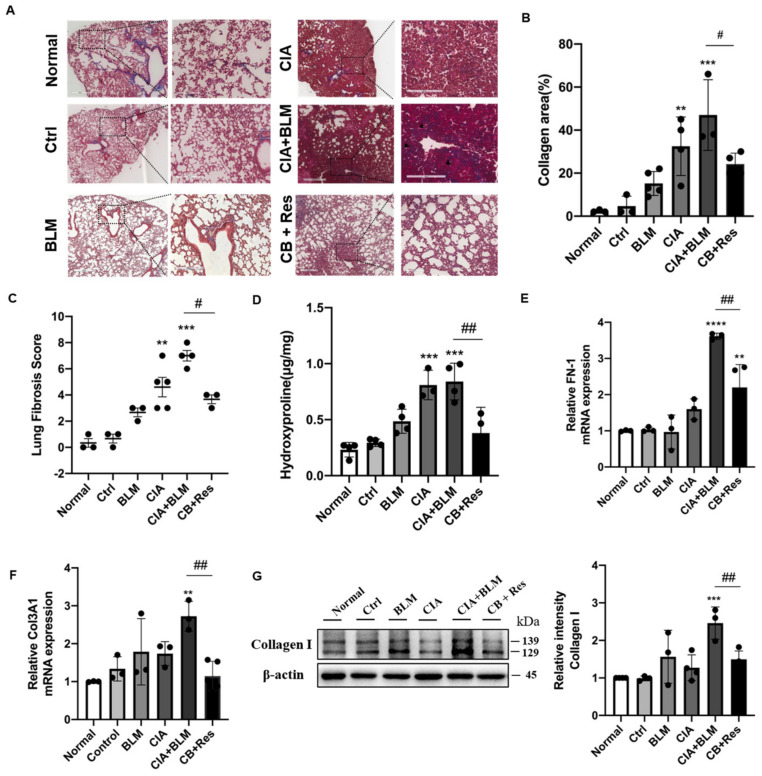
Resveratrol treatment attenuated lung fibrosis in CIA+BLM mice. (**A**) Representative image of pulmonary Masson’s staining shown at 40× (left panels) and 200× (right panels) magnification. Scale bar, 50 μm. Black arrows indicate the bands of fibrous tissues. (**B**) Quantitative analysis of the collagen area of each group. (**C**,**D**) Assessments of fibrosis in each group and levels of hydroxyproline in the lung were measured. (**E**,**F**) The mRNA levels of FN-1 and COL3A1 were quantified. (**G**) The levels of collagen I in the lungs were detected using immunoblotting and were quantified as described above. Values represent the mean ± SEM (n = 3–5). Compared with the control group, ** *p* < 0.01, *** *p* < 0.001 and **** *p* < 0.0001; compared with the model group (CIA+BLM), # *p* < 0.05 and ## *p* < 0.01.

**Figure 4 molecules-27-08475-f004:**
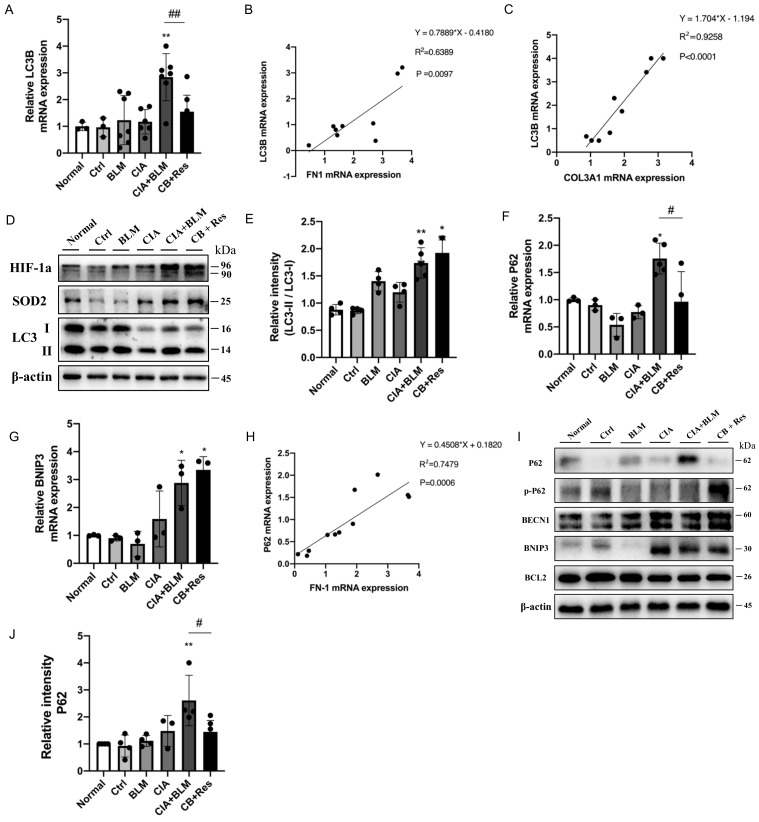
Resveratrol treatment reversed the accumulation of P62 to enhance autophagy levels and attenuate fibrosis. (**A**) The mRNA level of LC3B was quantified. (**B**,**C**) The correlation between LC3B and FN1 and COL3A1 gene expression showed a stronger positive correlation. Spearman’s r and *p* values are indicated. (**D**) Levels of SOD2, HIF-1a, and LC3 in lung tissues were detected using Western blotting. (**E**) The immunoblotting data of LC3 II/I were quantified as described above. (**F**,**G**) The mRNA levels of P62 and BNIP3 were quantified. (**H**) The correlation between P62 and FN1 gene expression showed a stronger positive correlation. Spearman’s r and *p* values are indicated. (**I**) Levels of P62, p-P62(phospho-SQSTM1/p62(Ser349)), BNIP3, BECN1, and BCL2 in lung tissues were detected using Western blotting. (**J**) The relative intensity of P62 was quantified as described above using Western blotting. Values represent the mean ± SEM (n = 3–5). * *p* < 0.05 and ** *p* < 0.01 compared with the control group, and compared with the model group (CIA+BLM), # *p* < 0.05 and ## *p* < 0.01.

**Figure 5 molecules-27-08475-f005:**
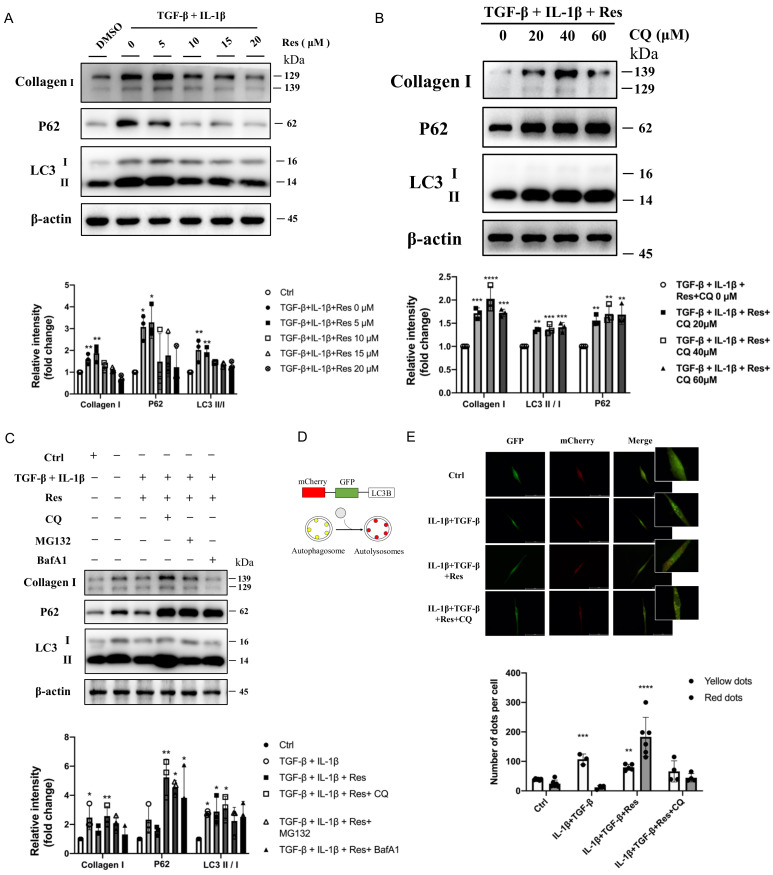
Resveratrol reversed disruption of autophagosome–lysosome fusion in vitro to improve fibrosis. (**A**) Immunoblotting of MRC-5 cells treated with TGF-β (10 ng/mL), IL-1β (5 ng/mL), and different concentrations of resveratrol were quantified as described above. (**B**) TGF-β+IL-1β, resveratrol, and CQ (0, 20, 40, 60 μM) treated MRC-5 cells for 24 h, and then they were immunoblotted and quantified as described above. (**C**) TGF-β (10 ng/mL) and IL-1β (5 ng/mL) induced MRC-5 cells treated with resveratrol (15 μM) were treated with different autophagy inhibitors (CQ, 40 μM; BafA1, 100 nM) and proteasome inhibitors (MG132, 10 μM), respectively, immunoblotted, and quantified as described above. (**D**) Diagram of mCherry-GFP-LC3B, an autophagic flux sensor. Autophagosomes appears yellow, whereas autolysosomes appear red, because GFP fluorescence is quenched under acidic environment after fusion with lysosomes. (**E**) mCherry-GFP-LC3B application to detect autophagy flow in MRC-5 cells before and after resveratrol or CQ treatment. Values represent the mean ± SEM (n = 3–5). Compared with the control group, * *p* < 0.05, ** *p* < 0.01, and *** *p* < 0.001 and **** *p* < 0.0001.

**Figure 6 molecules-27-08475-f006:**
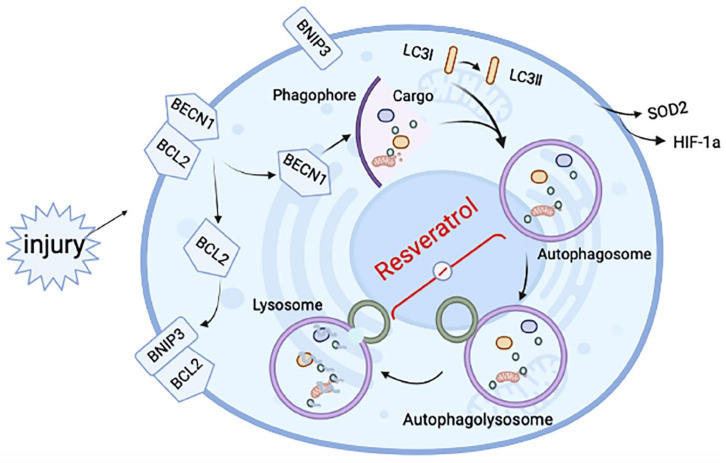
Schematic diagram of the mechanisms underlying resveratrol’s effect on induced RA-ILD. After tissue injury, hypoxia-related proteins change in the hypoxic microenvironment, making BNIP3 competitively bind BCL2 and release BCL2-bound BECN1, thus inducing autophagy. After autophagy is initiated, autophagosomes are formed, but the fusion with lysosomes to form autophagosomes is hindered. The effect of resveratrol is mainly in the fusion process of autophagosome and lysosome, to relieve the inhibition state and improve fibrosis.

**Table 1 molecules-27-08475-t001:** The gene-specific primers used in this study.

Gene	Primer Sequence (5′→3′)
GAPDH-F	TTCACCACCATGGAGAAGGC
GAPDH-R	GGCATGGACTGTGGTCATGA
COL3A1-F	CTGAAGATGTCGTTGATGTG
COL3A1-R	CTGATCCATATAGGCAATACTG
IL1β-F	CTGAACTCAACTGTGAAATGC
IL1β-R	TGATGTGCTGCTGCGAGA
TGF-β-F	ACAATTCCTGGCGTTACCTT
TGF-β-R	AGCCCTGTATTCCGTCTCC
BNIP3-F	TCCAGCCTCCGTCTCTATTT
BNIP3-R	CGACTTGACCAATCCCATATCC
LC3B-F	ATGCCGTCCGAGAAGACCTTCA
LC3B-R	CTGTGCCCATTCACCAGGAGGA
P62-F	GAACTCGCTATAAGTGCATGT
P62-R	AGAGAAGCTATCAGAGAGGTGG
FN-1-F	TGGTTTGGTCTGGGATCAATAG
FN-1-R	GTGACTTTCCTGCTCAAGGT

## Data Availability

Not applicable.

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
