# Peer review of "Resveratrol Ameliorates Fibrosis in Rheumatoid Arthritis-Associated Interstitial Lung Disease via the Autophagy–Lysosome Pathway"

_molecules, 2022, doi:10.3390/molecules27238475_

Round 1

Reviewer 1 Report

In this study, the authors made interesting discoveries that treatment with resveratrol can attenuate pulmonary fibrosis in a RA-ILD animal model and can also enhance autophagy activities in the lungs with conflict evidence. Inhibit autophagic activities in MRC-5 cells in vitro diminishes resveratrol’s effects in suppressing the expression of collagen.

Major concerns,

1.       The manuscript is very difficult to understand. The Methods and Materials section needs to be re-organized. It is not necessary to describe study design here, but simply list animal or experimental procedures. Please do use full names instead of abbreviations when mentioned for the first time. Missing inflammation score criteria or citations for joint or lung histology. The Results section also needs to be reworded. Table 1, primer sequences, belongs to M&M section.

2.       All the bar figures shall be switched to individual dot figures.

3.       Figure 1. Hydroxyproline assays are required to determine lung fibrosis. Histology and wet lung weight are not sufficient to make such a conclusion. 1D, the color of trichrome staining is strange. It is difficult to see blue color. Same problem with Figure 3A.

4.       Figure 2A, missing “II” after Col-; 2B needs measurements of foot volumes. 2E does not provide meaningful information, consider to be deleted; 2G, the y axis should be ”wet weight/dry weight”.

5.       Figure 4, the evidence of resveratrol’s effect on enhance autophagy seems not very solid. 4D, western blot shows that the CB+Res mice has very high levels of LC3 I and II, seems to be inconsistent with the decreased levels of RNA expression; 4E, the ratio of LC3 II/LC3 I is not increased by Res treatment. These issues need to be thoroughly discussed.

6.       Figure 5E, the resolution of the images is too low. Please make the red or yellow dots within each cell visible.

Reviewer 2 Report

The study "Resveratrol ameliorates fibrosis in rheumatoid arthritis-associated interstitial lung disease via autophagy-lysosome pathway" by Bao et al reveals the role of resveratrol in treating rheumatoid arthritis-associated interstitial lung disease. They showed that resveratrol promotes autophagy and therapeutic effects seem to be ceased when an autophagy inhibitor is present. The finding is interesting, but the data is not convincing enough in its present form. Additional control experiments are needed. Also, some of the data quality could be improved.

Major:

1.    Therapeutic effect of resveratrol could be reverted by CQ in cell model. How about animal model? (CQ could be delivered to mouse model too. E.g Vodicka, Petr, et al. (2014; https://pubmed.ncbi.nlm.nih.gov/25062859/) and Yi, Heqing, et al. (2018; https://pubmed.ncbi.nlm.nih.gov/29207150/)

2.    The background level of the western blots are generally very high. It may affect the quantification of the study. Also, some blots seem to have been cut very near to the desired molecular weight (e.g p62 of Fig 4i, there is a very obvious cut in the upper region).

3.    Also for the western blots, the authors should indicate the molecular weight.

4.    For western blots that show double bands, e.g. Collagen I, p62, etc. Author should indicate whether “both” or “only one” band are the correct band. It is also not sure which or both band are employed in quantification.

5.    Some bands are heavily overexposed. While the trends and patten could be seen, this dampens the reliability of the quantification. It is possible to show a lighter exposure especially when should then change of autophagy-related proteins?

6.    Why phosphorylated P62 is not quantified?

Minor:

1.    What is the rationale for only male mice being used in the study? Could it be explained in the methodology?

2.    Please arrange the sub-figures in a more ordered way. Now they seem a bit random. (For example, in Fig 4, Why A, B, E in the first row, and then C, D, F in the second row? )

Round 2

Reviewer 1 Report

The revision has dressed most of my concerns. However, I still have a few concerns in the Methods section.

1.       Were the mouse treatments approved by a local institutional animal care and use committee?

2.       Many abbreviations such as Ctrl, BLM, CIA, CIA+BLM, CB + Res, etc. are not explained.

3.       Numbers of mice per group are not consistent. It has been mentioned “5 mice in each group” and “5-8 in each group” in Methods. In the figures, there are many groups with 3 to 4 data points.

Author Response

TRANSLATE with x English

Arabic Hebrew Polish
Bulgarian Hindi Portuguese
Catalan Hmong Daw Romanian
Chinese Simplified Hungarian Russian
Chinese Traditional Indonesian Slovak
Czech Italian Slovenian
Danish Japanese Spanish
Dutch Klingon Swedish
English Korean Thai
Estonian Latvian Turkish
Finnish Lithuanian Ukrainian
French Malay Urdu
German Maltese Vietnamese
Greek Norwegian Welsh
Haitian Creole Persian  

TRANSLATE with COPY THE URL BELOW Back EMBED THE SNIPPET BELOW IN YOUR SITE Enable collaborative features and customize widget: Bing Webmaster Portal Back
